# Learning from the Past: Parametric Analysis of Cob Walls

Alejandro Jiménez Rios 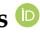

Department of Built Environment, Oslo Metropolitan University, 0130 Oslo, Norway;
alejand@oslomet.no; Tel.: +47-6723-8838

**Abstract:** In this paper, the results obtained from a series of parametric analyses, where the influence that geometric and mechanical parameters have in the structural response of existing vernacular cob walls within an Irish context, are presented. A design of experiments using central composite designs was implemented along with analysis of variance following two computational approaches, namely, the finite element method and kinematic limit analysis. As results, a series of response surfaces and parametric equations with which it is possible to compute safety factors and collapse multipliers (within the range of values studied) are provided. Based on the results obtained, it could be concluded that traditional cob walls in Ireland are very robust. Relatively high acceleration values, unlikely to happen in a low seismic hazard region such as Ireland, would be needed to start the collapse mechanisms studied or cause yielding in typical vernacular cob walls. Furthermore, the equations generated with the refined regression models can be used by practitioners as a first approach to estimate the safety levels of existing cob buildings with similar characteristics.

**Keywords:** cob; finite element method; limit analysis; design of experiments; response surface; earthen vernacular architecture; cultural heritage conservation





## 1. Introduction

The study of vernacular architecture is important as it has the potential to contribute to the global efforts to attain sustainable development as aimed for by the sustainability development goals adopted by the United Nations [1]. Of special interest is the study of earthen vernacular architecture, whose important role, for example, in the development of zero-energy in dry and desertic zones, as studied by Alrashed et al. [2], and in aspects such as energy consumption reduction and increased thermal comfort performance, as presented by Li et al. [3], has been highlighted. One of the main factors contributing to the sustainability of earthen vernacular architecture is the adaptation of the construction techniques to the availability of local resources [4].

Earthen vernacular architecture structures can be classified in different ways based on (i) the construction method type, as either dry or wet; (ii) the structural function, as load-bearing or non-load-bearing; and (iii) the structural element type, as monolithic, masonry, or infill. Examples of masonry earthen structures are adobe [5], compressed earth blocks [6], and sod/turf [7]. On the other hand, wattle-and-daub [8] and shot earth [9] are examples of infill earthen structures, whereas rammed earth [10] and cob [11] elements are normally classified as monolithic structural elements. Infill elements are commonly used as partition walls and act as non-load-bearing elements. On the contrary, both masonry and monolithic elements typically support their weight as well as other parts of the structure acting as load-bearing elements. As recognized by Khan et al. [12], masonry is a non-homogeneous anisotropic material and its correct characterization needs to account for the inherent properties of its units (adobes, compressed earth blocks, etc.) and joints (mortar or the lack of it). Conversely, earthen monolithic elements can be idealized as homogeneous, in analogy to how plain concrete is understood, and their characterization is performed based on the bulk properties of the material [13].

Even though cob has a nonlinear structural behavior [14], it presents an initial linear relationship between stress and strain up to a level of about 30% of its peak strength [13]. This indicates that if the loads acting on the structure do not cause stresses beyond the linear elastic range of the material, cob structures may be studied with a simplified linear analysis. Linear analyses are very attractive for practitioners and are used in the majority of practical designs as they are easier and faster to perform than nonlinear analyses. Thus, it is of interest to find out up to what point the different values of the parameters influencing the response of the structure will allow a simplified linear analysis to be performed with a high enough degree of accuracy.

The objective of this paper is to analyze existing cob vernacular buildings to provide simplified analysis tools for practitioners to better understand their structural response. To achieve this, the influence that geometric parameters and external actions have on the response of cob walls has been identified by means of a parametric analysis. Guidance is provided in the form of parametric equations capable of computing collapse multipliers (for the limit analysis approach) or safety factors (for the linear elastic finite element method (FEM) approach). Such guidance could as well be used as a tool for the initial dimensioning, and later design, of new sustainable and resilient cob buildings, which is a field that has recently received increased attention [15–19].

The paper aims to cover representative remaining earthen vernacular buildings in Ireland [11], which typically are single storied, regular in plan, longer than they are wide, and without rigid diaphragms to ensure a *"box-like"* structural behavior. The methodology and results presented in this work could as well be extrapolated for their application to buildings with similar characteristics elsewhere, in particular, to similar building typologies such as the ones present in the United Kingdom and the north of France [20]. Moreover, simplified analysis tools such as the one proposed in this paper are usually sought after to be implemented in vulnerability assessments of a large number of built assets located within a certain region. Vulnerability assessments of masonry building [5] and adobe building [21] typologies are available in the literature. On the other hand, to the extent of the knowledge of the author of this paper, no such study has been performed yet for cob vernacular buildings in Ireland or in other regions with similar building typologies. Thus, the work presented in this manuscript represents a novel contribution to the field as a first attempt to investigate the performance of this group of built assets.

The results presented in this paper consist of a series of response surfaces and parametric equations with which it is possible to compute safety factors and collapse multipliers (within the range of values studied). Based on those results, it is concluded that traditional cob walls in Ireland are very robust. Relatively high acceleration values, unlikely to happen in a low seismic hazard region such as Ireland, would be needed to start the collapse mechanisms studied or cause yielding in typical vernacular cob walls. Furthermore, the equations generated with the refined regression models can be used by practitioners as a first approach to estimate the safety levels of existing cob buildings with similar characteristics.

## 2. Materials and Methods

The methodology followed to perform the parametric analysis of cob walls consisted of three stages: parameters definition, FEM/macro-element kinematic limit analyses, and interpretation of the results (see Figure 1). Two analysis approaches were followed with the aim of exploring the pros and cons of each of them and providing alternatives to practitioners: linear elastic FEM analysis and macro-element kinematic limit analysis.

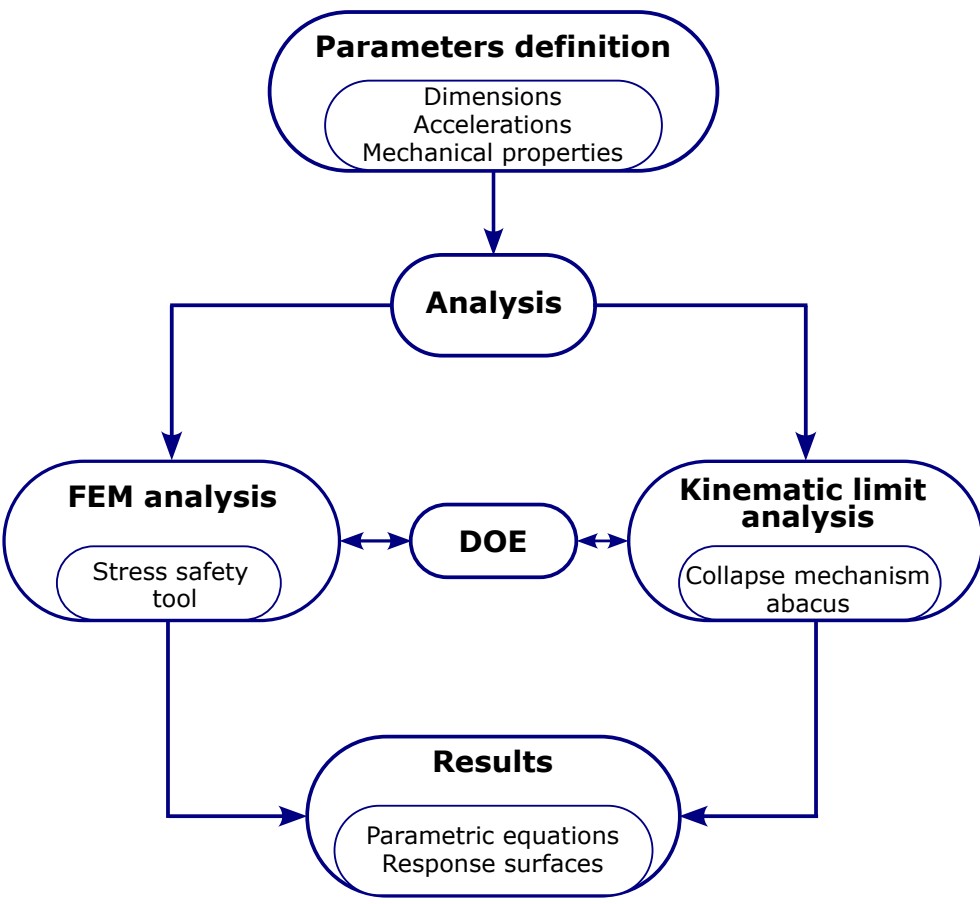

**Figure 1.** Multilevel approach for the simplified analysis of cob walls.

*2.1. Parameters*

The parameters studied were the walls' dimensions and mechanical properties, as well as the accelerations to which cob walls may be subjected during earthquakes. The *"best-guess"* fixed cob's mechanical properties' values used in this parametric analysis correspond with those reported in [13,22–24], as those are two of the most recent and complete studies performed to determine cob's properties. The values are summarized in Table 1.

**Table 1.** Cob wall's mechanical properties.

| Property | Value |
|---|---|
| Density (kg/m$^3$) | 1475 |
| Young's modulus (Pa) | $1.021 \times 10^9$ |
| Poisson's ratio (-) | 0.140 |
| Tension yield strength (Pa) | $0.062 \times 10^6$ |
| Compression yield strength (Pa) | $0.477 \times 10^6$ |

Currently, a National Inventory of Architectural Heritage (NIAH) which identifies, records, and evaluates the post-1700 architectural heritage in Ireland is available (https://www.buildingsofireland.ie/, accessed on 1 June 2023). Unfortunately, the NIAH is limited to providing an overall description of the buildings and a summarized appraisal of their value; it does not record technical information which may be of interest to their study. Dimensions of traditional earthen vernacular buildings in Ireland are only generally recorded by several authors [25–29] and unfortunately there is not a comprehensive database

containing such information. The geometric parameters adopted correspond to the values reported by such researchers based on studies of surviving cob buildings and records of historical cob buildings in Ireland. The minimum and maximum values of geometrical parameters adopted for this study are summarized in Table 2. The influence of wall voids, gables, chimneys, or the variability in the mechanical properties of cob in the structural behavior of the walls are not covered in this paper; this may be addressed in further work applying a more advanced and detailed analysis as suggested in [30].

**Table 2.** Cob wall's geometric parameter values.

| Parameter | Minimum Value | Maximum Value |
|:---:|:---:|:---:|
| Length (m) | 3.0 | 9.0 |
| Height (m) | 1.8 | 3.05 |
| Thickness (m) | 0.4 | 0.9 |

Ireland is a region characterized by low seismic activity. This fact may justify the use of simplified methods of analysis to describe cob buildings' structural behavior as it is presumed that the surviving vernacular cob buildings would not experience nonlinear behavior caused by lateral accelerations. The same can also be said for newly constructed cob buildings. The Seismic Hazard Harmonization in Europe (SHARE) project [31] produced a series of ground motion hazard maps for various spectral ordinates and exceedance probabilities in Europe. Figure 2 shows the peak ground acceleration (PGA, in g) map of Europe for a 10% exceedance probability in 50 years. The acceleration values for a low seismicity area, and in this case the type of area corresponding to Ireland, can go from 0 up to 0.10 g. The minimum and maximum values adopted for this parametric analysis were 0.05 g and 0.10 g (0.4905 m/s$^2$ and 0.981 m/s$^2$), respectively. Even though these acceleration values seem to be quite small, damage in earthen structures has been reported at similar intensities [32]. Furthermore, as it is intended that the methodology presented in this section could be extrapolated and applied in other regions where buildings with similar characteristics exist, i.e., the United Kingdom and the north of France, it was considered appropriate to explore the effect of earthquakes on cob structures.

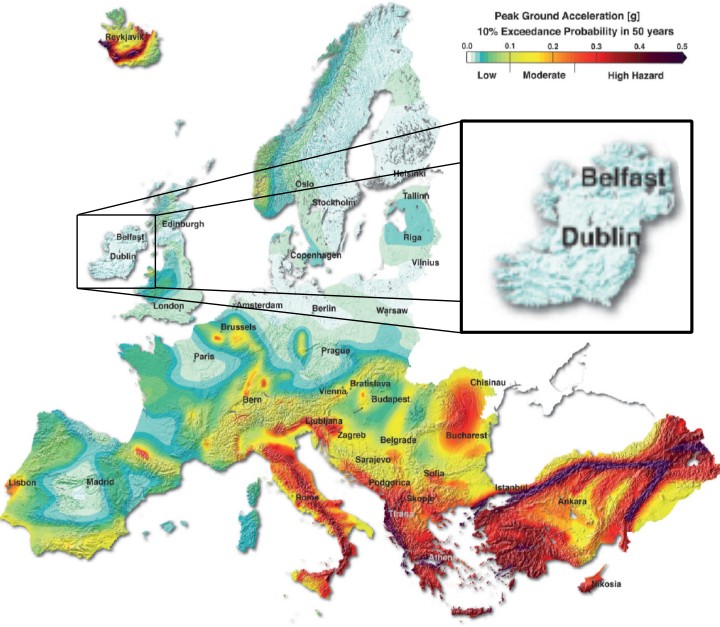

**Figure 2.** PGA (in g) map of Europe for a 10% exceedance probability in 50 years (adapted from [31]).

The simplified models studied in this parametric analysis are static. The dynamic accelerations caused by an earthquake in an existent cob structure were simplified into static inertial forces by using D'Alembert's principle. This principle establishes that *"a mass develops an inertial force proportional to its acceleration and opposing it"* [33].

### 2.2. Linear Elastic FEM Analysis

Linear elastic FEM analyses were performed using ANSYS [34]. It was decided to study the response of the walls subjected to both in-plane as well as out-of-plane behavior due to the multidirectional nature of dynamic actions. For either case, a free-standing cob wall fixed (translational degrees of freedom constrained) at its bottom was simulated. This assumption is justified by the fact that most earthen vernacular buildings in Ireland do not have rigid diaphragm roofs (lack of bond beam, roof trusses directly placed on top of cob wall). Thus, cob walls do not present a *"box-like"* behavior and most of the time act individually to support the actions imposed upon them. Moreover, vertical cracks are quite common, especially at the corners (see Figure 3), which would cause the detachment of perpendicular wall connections easing their out-of-plane overturning under horizontal loads.

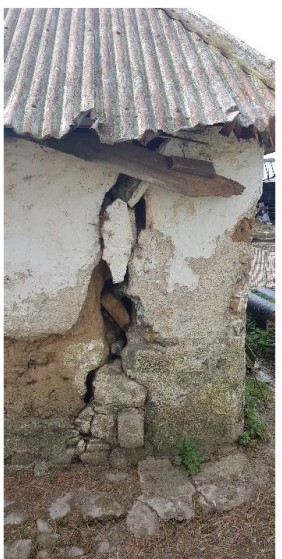
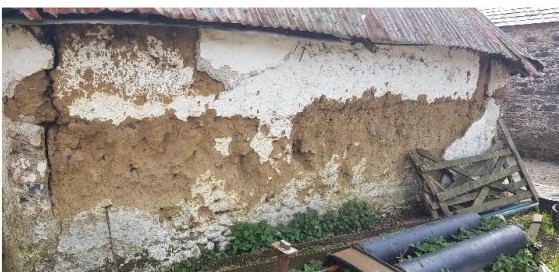

**Figure 3.** Vertical cracks in a cob building located in County Kildare, Ireland (pay special attention to the cracks at the corners running all along the wall's height).

These assumptions were adopted to simulate a general *"worst-case scenario"* in which a cob wall, part of a surviving cob building in Ireland, may be found. When dealing with existing structures that have cultural heritage value, a detailed study of the structure is desirable to avoid the design of over conservative interventions which may endanger the historic fabric [35]. Therefore, the assumptions adopted for specific case studies must be based on the results provided by a structural integrity inspection after which the following may be found:

- wall interconnectivity (perpendicular walls providing out-of-plane support);
- top restraint at roof level;
- *"box-like"* behavior (provided by a strengthening/retrofitting intervention).

Any of these factors would improve the structural response of the building and if they are neglected, the design of the intervention may be too invasive. On the other hand, wall voids and the presence of damage or decay would diminish the structure's capacity.

The walls were subjected to their weight, to an external force at their top, simulated as a concentrated mass representing the loads transferred by the roof, and finally, to a horizontal acceleration representing the effects of dynamic loads on the wall. The point

mass at the top of the wall was assumed to be that of a thatch roof with a thickness of 305 mm (including battens) and a self-weight of 450 N/m$^2$ [36], with a tributary span of 3 m giving a value of 137.6 kg/m for every meter of wall length. The setup of the in-plane and of the out-of-plane models are shown in Figure 4a,b, respectively. The finite element used was the SOLID186 [37] and the constitutive material model adopted was a simple linear elastic model.

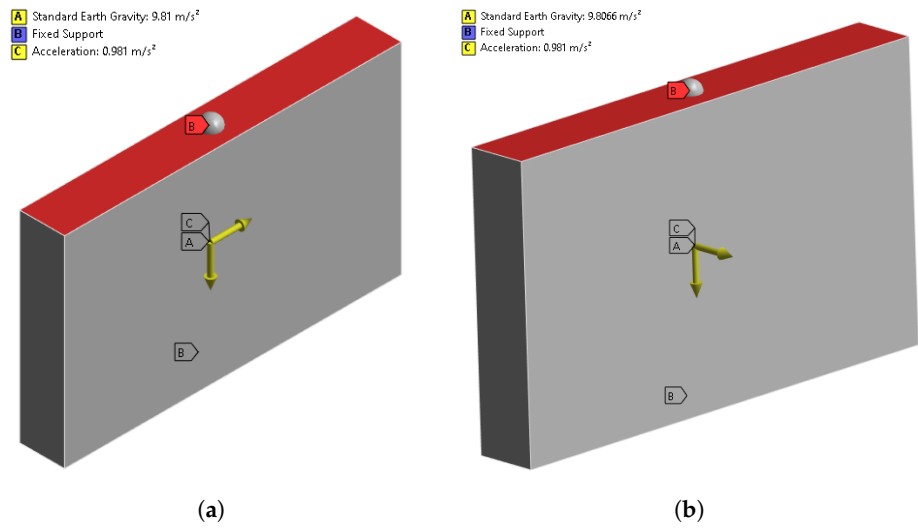

(**a**)            (**b**)

**Figure 4.** FEM model setup for (**a**) in-plane analysis and (**b**) out-of-plane analysis.

The response of the structure was monitored with the help of a Mohr–Coulomb stress safety tool. This tool is based on the Mohr–Coulomb theory for brittle materials and uses the principle stress values to determine the occurrence of failure. The yielding safety factor capacity of the material, $F_{ys}$, is computed with Equation (1) [38].

$$F_{ys} = (\sigma_1 / S_{yt} + \sigma_3 / S_{yc})^{-1}, \tag{1}$$

where $\sigma_1$ and $\sigma_3$ are the maximum and minimum principal stresses, and $S_{yt}$ and $S_{yc}$ are the material yielding tensile and compressive strengths.

Strictly speaking, the tool provides a distribution of safety factors throughout the model as it bases its calculations on the independent distributions of maximum and minimum stresses and not on the absolute principal stresses, developed most likely at two different locations in the model. Therefore, the minimum value of such a distribution is reported as the minimum factor of safety. The compressive strength of common brittle materials is usually much greater than their tensile strength. In this analysis, this assumption is considered to be valid for cob as well. It is assumed that cob's tensile ultimate strength corresponds to only 13% of its compressive ultimate strength [39]. The Mohr–Coulomb stress safety tool takes direct account of this theory and is often considered to provide conservative results.

### 2.3. Macro-Element Kinematic Limit Analysis

A second method of analysis was implemented to compare the results obtained with the simplified FEM analysis. This method is based on the abacus of collapse mechanisms reported by the new integrated-knowledge-based approaches to the protection of cultural heritage from the earthquake-induced risk (NIKER) European project [40]. The abacus of collapse mechanisms basically classifies the collapse mechanisms into two categories: in-plane and out-of-plane mechanisms. The identified in-plane damage caused by horizontal loads consist mainly in rocking, sliding shear failure, and diagonal cracking. On the other hand, the out-of-plane damage is mainly characterized by the development of vertical cracks. Vertical cracks may either start from the bottom or the top of the wall and some run

through the entire height of the wall. This causes the separation of important segments of the wall and can lead to the overturning of the element.

The macro-element kinematic limit analysis consists basically in finding the equilibrium condition under which, for a certain value of the $\alpha$ coefficient, the studied collapse mechanism will form [41–44]. Thus, $\alpha$ is defined as the seismic mass multiplier that leads to the formation of the collapse mechanism assumed and eventually to the element failure [45]. $\alpha$ is computed as the ratio between $a$, the ground acceleration, and $g$, the acceleration due to gravity.

The out-of-plane mechanism considered in this parametric analysis is presented in Figure 5. The thickness of the wall is represented by $T_{wall}$, its height by $H_{wall}$, and its length by $L_{wall}$. It is assumed that cob has zero tensile strength and that no internal sliding would happen, in other words, the walls behave as rigid bodies and rotate while supported in two idealized pined supports. For the case shown in Figure 5a, it was assumed that cob has infinite compressive strength and the bottom support is located at the edge of the wall. On the other hand, for the case presented in Figure 5b, the compressive strength of the material, $f_c$, is taken into account and the bottom support is located at a distance $t$ from the edge of the wall (this is a conservative assumption, a more realistic one would consider the location of the support at a distance of $t/3$ from the edge of the wall). The force $P$ represents the self-weight of the wall and the force $N$ the external loads transferred from the roof. To compute $N$, a thatch roof with a thickness of 305 mm (including battens) and a self-weight of 450 N/m$^2$ [36] has been considered with a tributary span of 3 m giving a value of 1350 N/m for every meter of wall length.

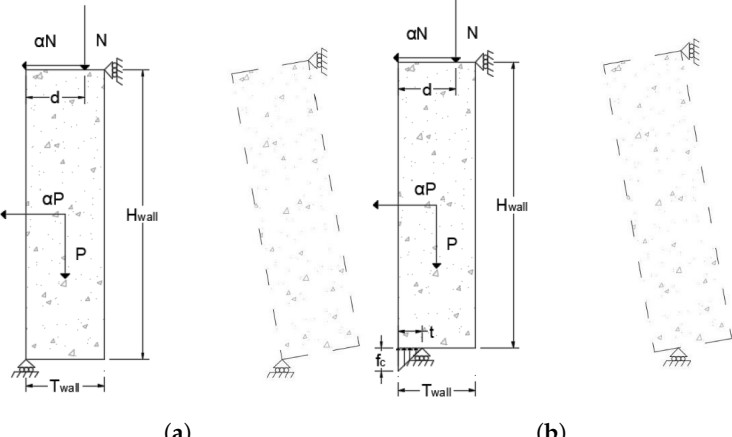

(**a**)             (**b**)

**Figure 5.** Out-of-plane mechanisms (**a**) assuming infinite compressive strength, and (**b**) accounting for the compressive strength of the material, $f_c$.

The equilibrium condition of the rigid body for the case is presented in Figure 5a, from where the expression to compute the $\alpha$ coefficient is obtained (see Equation (2)). It is worth noting that for this parameter analysis, the distance $d$ was assumed to be half of the wall thickness, thus, placing the external loads over the centroid of the wall.

$$\alpha = \frac{P(T_{wall}/2) + Nd}{P(H_{wall}/2) + NH_{wall}}, \tag{2}$$

To find the equilibrium condition of the rigid body shown in Figure 5b, it is necessary to find the value of $t$ at which the support is located from the edge of the wall. This distance is obtained by equating the vertical forces acting on the wall with the force developed at the support as a function of the material's compressive strength, $f_c$. Thus, for a unitary segment of wall (1 m length):

$$t = \frac{2(P + N)}{f_c * 1}, \tag{3}$$

The seismic mass multiplier is calculated using Equation (4).

$$\alpha = \frac{P(T_{wall}/2 - t) + N(d - T_{wall})}{P(H_{wall}/2) + NH_{wall}},$$
(4)

The in-plane mechanism considered in this parametric analysis is presented in Figure 6. It is the result of diagonal cracking on the cob wall. For the case shown in Figure 6a, it was assumed that cob has infinite compressive strength (along with zero tensile strength and no internal sliding) and the bottom support is located at the edge of the wall whereas that for the case presented in Figure 6b, the compressive strength of the material, $f_c$, is taken into account and the bottom support is located at a distance $t$ from the edge of the wall (this is a conservative assumption, a more realistic one would consider the location of the support at a distance of $t/3$ from the edge of the wall). For the in-plane analysis the distance $d$ was considered to be equal to half the distance between the wall supports.

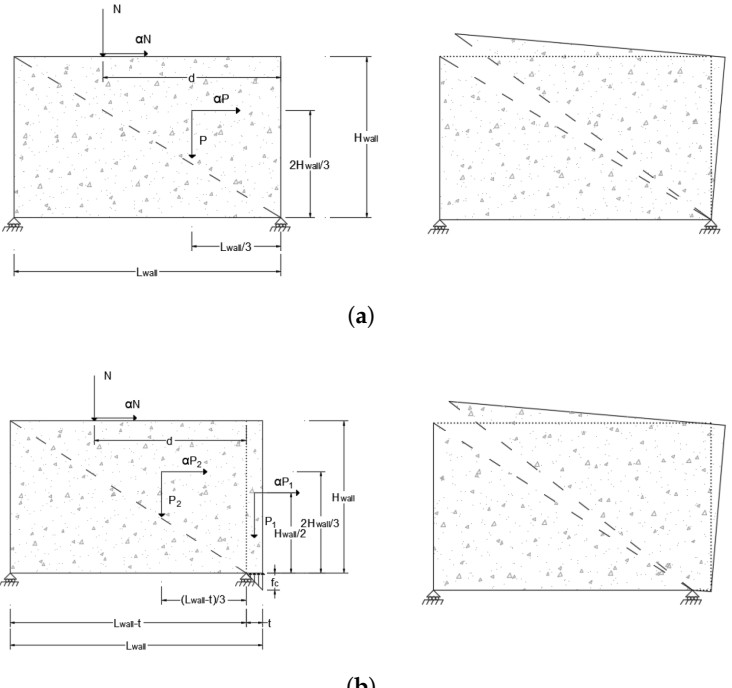

**(a)**

**(b)**

**Figure 6.** In-plane mechanisms (**a**) assuming infinite compressive strength, and (**b**) accounting for the compressive strength of the material, $f_c$.

The formulas used to calculate the seismic mass multiplier coefficient for case (a) and case (b) are presented in Equations (5) and (7), respectively. The value of $t$ was calculated using Equation (6). These equations were obtained by applying equilibrium equations of the vertical stabilizing forces, $P$ and $N$, and the horizontal destabilizing forces, $\alpha P$ and $\alpha N$.

$$\alpha = \frac{Nd + P(L_{wall}/3)}{NH_{wall} + P(2H_{wall}/3)},$$
(5)

$$t = \frac{P_2 + N}{\frac{T_{wall}f_c}{2} - H_{wall}T_{wall}\gamma_{cob}g + \frac{H_{wall}T_{wall}\gamma_{cob}g}{2}},$$
(6)

$$\alpha = \frac{Nd + P_2\frac{L_{wall} - t}{3} - P_1(t/2)}{NH_{wall} + P_2\frac{2H_{wall}}{3} + P_1(H_{wall}/2)},$$
(7)

where $\gamma_{cob}$ represents the cob's density. The values of $\alpha$ for the four mechanisms presented were computed using MATLAB [46] scripts.

*2.4. Design of Experiment (DOE)*

A DOE technique was used for both the linear elastic FEM analysis and the macro-element kinematic limit analysis to obtain meaningful results and, at the same time, to reduce the computational effort required. The DOE for the linear elastic FEM analysis was performed directly in ANSYS using the design exploration module [47], whereas the DOE for the macro-element kinematic limit analysis was performed using the statistics software Minitab [48].

A central composite design (CCD) method was implemented with the objective of studying the influence of the input parameters (geometric and accelerations for the FEM models and geometric for the limit analysis models) in the responses of the models ($F_{ys}$ for the FEM models and seismic mass multiplier for the limit analysis models). A full CCD was selected for the design type as this method is available as the default option in both applications and is the most commonly used method for the DOE to create response surfaces [49]. To ensure rotatability, adequate values for the distance of the star points were selected based on the relationship $\alpha_{CCD} = F_{CCD}^{1/4}$, where $\alpha_{CCD}$ is the axial spacing of the design points and $F_{CCD}$ is the number of points in the factorial part of the design, usually $F_{CCD} = 2^k$, where $k$ is the number of factors.

Finally, the design points generated with the CCD were used to build the response surfaces by applying a full 2nd-order polynomial algorithm. Then, the main effect and interaction effect plots were analyzed and a second response surface was created taking into account only those statistically significant terms ($p$-value $\leq 0.05$) to simplify the regression model. More details about the CCD method can be consulted elsewhere [50].

For both FEM and macro-element models the mechanical properties of cob were maintained constant. The wall geometric parameters and the levels of acceleration were used as input parameters for the FEM models in the DOE and the yield and ultimate safety factors computed were used as output parameters. On the other hand, only the geometric parameters were used as input parameters for the macro-element models and the computed $\alpha$ coefficients were the correspondent output parameters.

*2.5. Study Limitations*

The main limitations of the current study are related to the inherent numerical modeling simplifications adopted and to the uncertainties arising from the selection of parameter values due to the scarce availability of such information. When a simplified linear elastic analysis of an existing cob structure cannot be justified, either because future seismic events, high loading levels, or other conditions may be expected to cause a nonlinear response of the structure, the use of more advanced numerical modeling tools, such as the ones discussed in [51], is recommended.

Unfortunately, cob buildings have not received as much attention as other construction typologies, i.e., adobe or masonry, and as discussed in this section, the values of the parameters selected are based on limited experimental laboratory campaigns and registries in historical sources. Although the parametric equations proposed in this work have been derived using well-known and accepted principles [52,53], for a robust and strict validation further work is required. In Appendix A, a sensitivity analysis is presented to better understand how the uncertainties of key material parameters influence the parametric equations proposed in this work and as a first step forward toward their validation.

## 3. Results and Discussions

A free-standing wall fixed at its base and subjected to in-plane forces may be idealized as a plane-stress problem where the thickness of the wall could be neglected without significantly affecting the results. On the other hand, the same wall subjected to out-of-plane forces could be idealized as a plane-strain problem, and in this case, the length of the wall would not affect the results. The results of the parametric analysis performed have

confirmed these simple, but important, concepts and will be described in more detail in this section.

### 3.1. FEM Out-of-Plane Analysis

The DOE using a CCD rotatability design with four factors (length, height, thickness, and acceleration) generated 49 design points. The order in which the design points were generated by the CCD was random to avoid the effect of any nuisance variable in the output parameters (randomization is of paramount importance in the design of physical experiments, but it does not affect the numerical parametric analysis performed in this work).

The minimum value computed for $F_{ys}$ was equal to 3.0. This value was obtained for a wall with 4.5 m length, 2.74 m height, 0.53 m thickness, and subjected to a lateral acceleration of 0.86 m/s$^2$. The full quadratic model obtained through the DOE was refined by removing the non-significant terms. By taking into account only those statistically significant terms previously discussed, the expression shown in Equation (8) was obtained. Equation (8) can be used to estimate the value of $F_{ys}$ for every combination of wall height ($H_{wall}$ in m), thickness ($T_{wall}$ in m) and lateral acceleration ($a$ in m/s$^2$) within the range of values explored by the analysis presented in this paper. In this equation, the length of the wall is not included, as its effect on the out-of-plane response is non-significant, and this confirms the comment made at the beginning of the section regarding the fact that the length of the wall was expected not to influence the out-of-plane response of the wall.

$$F_{ys} = 19.51 - 9.55H_{wall} + 11.76T_{wall} - 7.41a + 1.33H_{wall}^2 - 9.9T_{wall}^2 + 6.34T_{wall}a, \quad (8)$$

If Equation (8) is evaluated for a wall height of 3.05 m, a thickness of 0.40 m, and an acceleration of 0.981 m/s$^2$, then, $F_{ys} = 1.1$. This indicates that even under the most adverse combination of values for the parameters explored, and based on the assumptions adopted, the material would not reach its yield point. Equation (8) was also used to create the response surfaces shown in Figure 7. Each one of the response surfaces shows information regarding the effect of two input parameters, while the third one is kept constant at its most critical value, on the response, $F_{ys}$. These response surfaces reflect the quadratic effect that the input parameters have on the response.

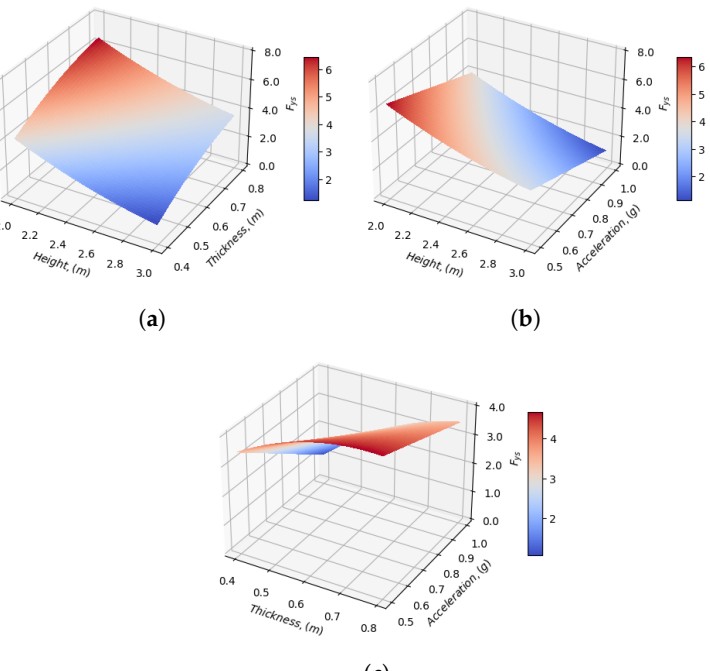

**Figure 7.** Response surfaces for $F_{ys}$ for the out-of-plane FEM model: (**a**) acceleration fixed at 0.981 g; (**b**) thickness fixed at 0.4 m; and (**c**) height fixed at 3.05 m.

### 3.2. FEM In-Plane Analysis

The DOE using a CCD rotatability design with four factors (length, height, thickness, and acceleration) generated 49 design points. The minimum value computed for $F_{ys}$ was equal to 5.52. This value was obtained for a wall with 6 m length, 3.05 m height, 0.65 m thickness, and subjected to an acceleration of 0.74 m/s². It would be expected that the length of the wall would play an important role in the wall in-plane response. Bulky walls tend to fail by shear, whereas slender walls tend to top up or fail by crushing at their base. Nevertheless, as the boundary conditions adopted prevented the in-plane rotation of the wall and, since the levels of stress created by the low levels of acceleration adopted did not exceed the yielding point of the material, the parametric analysis did not capture properly the effects of wall length.

The full quadratic model was refined by removing the non-significant terms. By taking into account only those statistically significant terms, the expression shown in Equation (9) was obtained. Equation (9) can be used to estimate the value of $F_{ys}$ for every combination of wall height ($H_{wall}$ in m) and lateral acceleration ($a$ in m/s²) within the range of values explored by the analysis presented in this paper. In this equation, the length and the thickness of the wall are not included, as their effect on the response is non-significant, and this confirms the comment made at the beginning of the section regarding the fact that the thickness of the wall was expected not to influence the in-plane response of the wall.

$$F_{ys} = 24.799 - 10.239 H_{wall} - 1.798a + 1.439 H_{wall}^2, \tag{9}$$

If Equation (9) is evaluated for a wall height of 3.05 $m$ and an acceleration of 0.981 m/s², then, $F_{ys} = 5.19$. This indicates that even under the most adverse combination of values for the parameters explored, and based on the assumptions adopted, the material would not reach its yield point. Furthermore, it can be seen that the out-of-plane model is more critical than the in-plane model as the critical value of $F_{ys}$ obtained for the out-of-plane model was only 1.10. Equation (9) was also used to create the response surface shown in Figure 8. This response surface shows how $F_{ys}$ varies based on the values of height and acceleration. This response surface reflects the quadratic effect that the input parameters have on the response

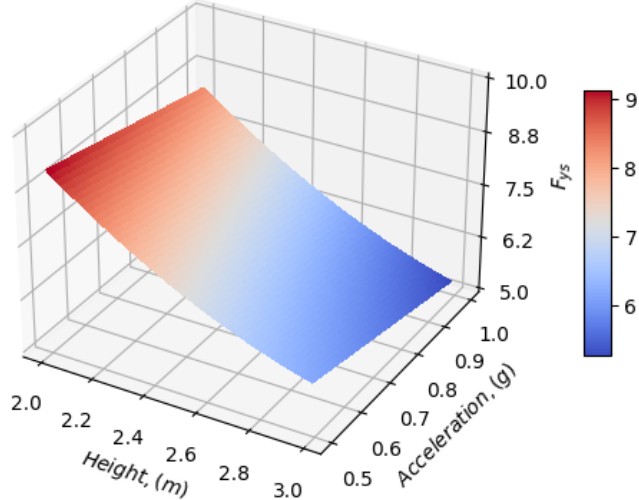

**Figure 8.** Response surface for $F_{ys}$ for the in-plane FEM model.

### 3.3. Macro-Element Out-of-Plane Analysis Assuming Infinite Compressive Strength

The design points generated with Minitab using a CCD with three factors (length, height, and thickness) consisted of eight cube points, six center cube points, and six axial points. A value for the distance of the axial points equal to 1.68179 was used to ensure rotatability, and 20 design points were generated.

The minimum value computed for $\alpha$, the seismic mass multiplier, was equal to 0.08. This value was obtained for a wall with 6.0 m length, 2.43 m height, and 0.23 m thickness (a thickness of 0.23 m is outside the range of values adopted, it is automatically generated by dividing the minimum value of 0.4 m by the specified value $a_{CCD} = 1.68179$ for the axial spacing). A mass seismic multiplier of 0.08 corresponds to a lateral acceleration of 0.7848 m/s$^2$ or to a PGA of 0.08 g. This level of acceleration corresponds to the upper range of values for low-seismic-hazard areas (see Figure 2).

As previously discussed, it is expected that the higher and the thinner a wall is, the more critical its safety is when subjected to out-of-plane forces. This assumption is verified by the data generated with this parametric macro-element out-of-plane analysis. The non-influence of wall length in the values computed for the seismic mass multiplier, as would be the case in a plane-strain simplified model, is explained by the fact that this parameter is not part of Equation (4).

By taking into account only those statistically significant terms, the expression shown in Equation (10) was obtained. Equation (10) can be used to estimate the value of $\alpha$ for every combination of wall height ($H_{wall}$ in m) and thickness ($T_{wall}$ in m) within the range of values explored by the analysis presented in this paper. In this equation, the length of the wall is not included, as its effect on the response is non-significant, and this is in agreement with the comment made at the beginning of the section regarding the fact that the length of the wall was expected not to influence the out-of-plane response of the wall.

$$\alpha = 0.238 - 0.220 H_{wall} + 0.859 T_{wall} + 0.046 H_{wall}^2 - 0.178 H_{wall} T_{wall}, \tag{10}$$

Equation (10) was also used to create the response surface shown in Figure 9. The response surface shows information regarding the effect of the two input parameters wall height and thickness on the response, $\alpha$. This response surface reflects the quadratic effect the input parameters have on the response.

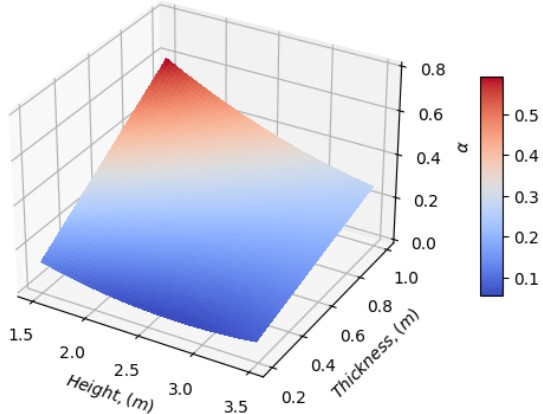

**Figure 9.** Response surface for the out-of-plane mechanism assuming infinite compressive strength.

### 3.4. Macro-Element Out-of-Plane Analysis Taking into Account the Compressive Strength of the Material

The compressive strength of the material was introduced as a categorical parameter with two levels representing both the yield and the ultimate compressive strength of cob (see Table 1). Therefore, the DOE using a CCD with two continuous factors (height and thickness) and one categorical factor (compressive strength) consisted of eight cube points, ten center cube points, and eight axial points. A value for the distance of the axial points equal to 1.41421 was used to ensure rotatability, and 26 design points were generated.

The minimum value computed for $\alpha$ was equal to 0.07. This value was obtained for two parameter combinations: a wall with 2.43 m height, 0.30 m thickness (a thickness of 0.30 m is outside the range of values adopted, it is automatically generated by dividing the minimum value of 0.4 m by the specified value $a_{CDD} = 1.41421$ for the axial spacing),

and a compressive strength of 0.48 MPa; and for a wall with 3.05 m height, 0.4 m thickness, and a compressive strength of 0.48 MPa. A mass seismic multiplier of 0.07 corresponds to a lateral acceleration of 0.6867 m/s$^2$ or to a PGA of 0.07 *g*. This level of acceleration corresponds to the upper range of values for low-seismic-hazard areas (see Figure 2).

The full quadratic model was refined by removing those non-significant terms. Thus, the expression shown in Equation (11) was obtained for a compressive strength of 0.48 MPa and the expression shown in Equation (12) was obtained for a compressive strength of 1.59 MPa. Equations (11) and (12) can be used to estimate the value of $\alpha$ for every combination of wall height ($H_{wall}$ in *m*) and thickness ($T_{wall}$ in *m*) within the range of values explored by the analysis presented in this paper based on the different compressive strengths mentioned. In these equations, the length of the wall is not included, as its effect on the response is non-significant, and this is in agreement with the comment made at the beginning of the section regarding the fact that the length of the wall was expected not to influence the out-of-plane response of the wall.

$$\alpha_{0.48} = 0.280 - 0.288H_{wall} + 0.778T_{wall} + 0.063H_{wall}^2 - 0.179H_{wall}T_{wall}, \tag{11}$$

$$\alpha_{1.59} = 0.330 - 0.288H_{wall} + 0.778T_{wall} + 0.063H_{wall}^2 - 0.179H_{wall}T_{wall}, \tag{12}$$

If Equations (11) and (12) are evaluated for a wall height of 3.05 m and a thickness of 0.40 m, then, $\alpha_{0.48} = 0.080$ and $\alpha_{1.59} = 0.130$. This indicates that under the most adverse combination of values for the geometric parameters explored, and based on the assumptions adopted, a cob wall would have to be subjected to an acceleration of 0.913 m/s$^2$ or 0.09 g to generate the out-of-plane mechanism if the yield compressive strength value is taken into account, and would have to be subjected to an acceleration of 1.196 m/s$^2$ or 0.1196 g to generate the out-of-plane mechanism if the ultimate compressive strength value is taken into account. The decision as to what equation should be used would depend on the degree of damage that is considered acceptable for the structure. The yield of the materials could be adopted for a serviceability limit state (SLS), whereas the ultimate strength of the materials would be adequate for an ultimate limit state (ULS) scenario.

Nevertheless, as this level of acceleration corresponds to the lower range of values for a moderate-seismic-hazard area, and since Ireland is located in a low-seismic-hazard area, it can be concluded that the out-of-plane mechanism analyzed would not form in Irish surviving cob walls. On the other hand, this level of acceleration can be expected to affect similar cob buildings in other regions, such as the United Kingdom and the north of France, and the methodology presented in this section could be implemented to evaluate the safety level of those buildings. Equations (11) and (12) were also used to create the response surfaces shown in Figure 10 for values of compressive strength of 0.48 MPa and 1.59 MPa, respectively. The response surface shows information regarding the effect of two input parameters, wall height and thickness, on the response, $\alpha$.

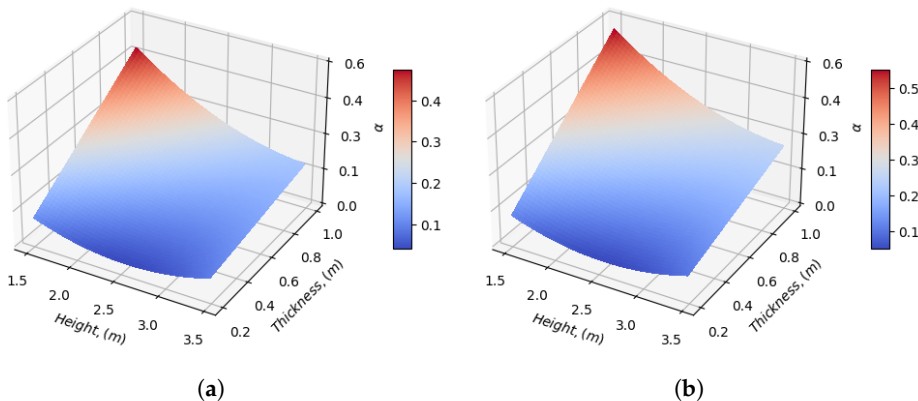

(**a**)             (**b**)

**Figure 10.** Response surface for the out-of-plane mechanism for a compressive strength of cob of (**a**) 0.48 MPa and (**b**) 1.59 MPa.

### 3.5. Macro-Element In-Plane Analysis Assuming Infinite Compressive Strength

The design points generated with Minitab using a CCD with three factors (length, height, and thickness) consisted of eight cube points, six center cube points, and six axial points. It used a value for the distance of the axial points equal to 1.68179 to ensure rotatability, and generated 20 design points.

The minimum value computed for $\alpha$ is equal to 0.20. This value was obtained for a wall with 0.95 m length, 2.43 m height, and 0.65 m thickness (a length of 0.95 m is outside the range of values adopted, it is automatically generated by using the axial spacing value adopted). A mass seismic multiplier of 0.20 corresponds to a lateral acceleration of 1.96 m/s$^2$ or to a PGA of 0.2 g. This level of acceleration corresponds to the upper range values for moderate-seismic-hazard areas.

As previously discussed, it is expected that the higher a wall is, the more critical its safety is when subjected to in-plane forces. The length of the wall is also expected to influence the failure of the wall subjected to in-plane forces; bulky walls tend to fail by shear, whereas slender walls tend to rock and fail by crushing at their base. The effects of both parameters on the wall's response are verified by the data generated with this parametric macro-element in-plane analysis. The non-influence of wall thickness in the values computed for the seismic mass multiplier, as would be the case for a plane-stress simplified problem, is explained by the fact that this parameter is not part of Equation (5).

One of the assumptions of the macro-element limit analysis consists of the neglect of internal sliding of the element, in other words, no shear failure can be detected with this simplified model. If the length of the wall increases, it is expected that its rotation, the mechanism that is being studied, would become less critical and a higher acceleration would be needed to initiate it but, in reality, this would not mean that the wall is safe from shearing failure. The lack of capability to detect that type of failure is one of the main shortcomings of the simplified model adopted in this parametric analysis.

By taking into account only those statistically significant terms, the expression shown in Equation (13) was obtained. Equation (13) can be used to estimate the value of $\alpha$ for every combination of wall height ($H_{wall}$ in m) and length ($L_{wall}$ in m) within the range of values explored by the analysis presented in this work. In this equation, the thickness of the wall is not included, as its effect on the response is non-significant, and this is in agreement with the comment made at the beginning of the section regarding the fact that the thickness of the wall was expected not to influence the in-plane response of the wall.

$$\alpha = 1.516 + 0.436 L_{wall} - 1.262 H_{wall} + 0.253 H_{wall}^2 - 0.091 L_{wall} H_{wall}, \tag{13}$$

If Equation (13) is evaluated for a wall height of 3.05 m and a length of 3.00 m, then, $\alpha = 0.4968$. This indicates that under the most adverse combination of values for the geometric parameters explored, and based on the assumptions adopted, a cob wall would have to be subjected to an acceleration of 4.968 m/s$^2$ or 0.4968 g so that the in-plane mechanism studied is generated. This level of acceleration corresponds to the upper range of values for a high-seismic-hazard area. As Ireland is located in a low-seismic-hazard area, it can be concluded that the in-plane mechanism analyzed would not form in Irish surviving cob walls. Equation (13) was also used to create the response surface shown in Figure 11. The response surface shows information regarding the effect of two input parameters, wall height and length, on the response, $\alpha$.

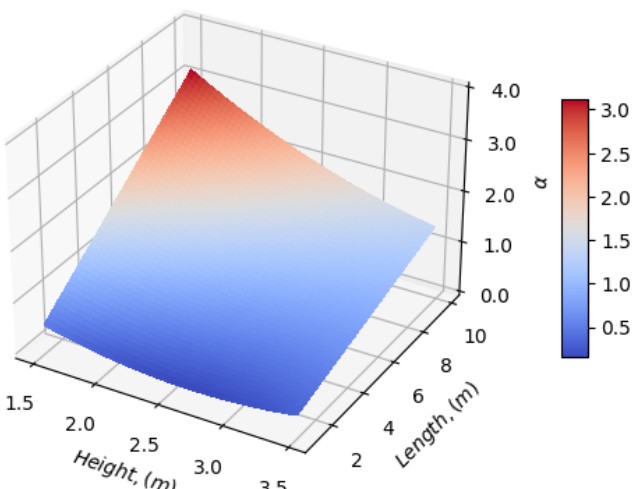

**Figure 11.** Response surface for the in-plane mechanism assuming infinite compressive strength.

*3.6. Macro-Element In-Plane Analysis Taking into Account the Compressive Strength of the Material*

As it was observed in the parametric analysis of the previous sub-section that the thickness of the wall does not influence its in-plane response (the results of that parametric analysis confirmed the expected behavior based on the plane stress assumption), it was decided to ignore that parameter for the macro-element in-plane analysis taking into account the compressive strength of the materials. The compressive strength of the material was introduced as a categorical parameter with two levels representing both the yield and the ultimate compressive strength of cob (see Table 1). Therefore, the DOE using a CCD with two continuous factors (height and length) and one categorical factor (compressive strength) consisted of eight cube points, ten center cube points, and eight axial points. It used a value for the distance of the axial points equal to 1.41421 to ensure rotatability, and generated 26 design points.

The minimum value computed for $\alpha$ is equal to 0.31. This value was obtained for a wall with 2.43 m height, 1.76 m length (a length of 1.76 m is outside the range of values adopted, it is automatically generated by using the specified axial spacing value), and a compressive strength of 0.48 MPa. A mass seismic multiplier of 0.31 corresponds to a lateral acceleration of 3.041 m/s$^2$ or to a PGA of 0.31 g. This level of acceleration corresponds to the lower range of values for high-seismic-hazard areas. As the higher the compressive strength of the material, the smaller the value of $t$, it was expected that the response would also be influenced by this parameter. Effectively, a directly proportional relationship exists between cob's compressive strength and $\alpha$.

The full quadratic model was refined by removing those non-significant terms. Thus, the expression shown in Equation (14) was obtained for a compressive strength of 0.48 MPa and the expression shown in Equation (15) was obtained for a compressive strength of 1.59 MPa.

$$\alpha_{0.48} = 1.328 + 0.414 L_{wall} - 1.169 H_{wall} + 0.237 H_{wall}^2 - 0.091 L_{wall} H_{wall}, \tag{14}$$

$$\alpha_{1.59} = 1.468 + 0.414 L_{wall} - 1.169 H_{wall} + 0.237 H_{wall}^2 - 0.091 L_{wall} H_{wall}, \tag{15}$$

If Equations (14) and (15) are evaluated for a wall height of 3.05 m and a length of 3.00 m, then, $\alpha_{0.48} = 0.377$ and $\alpha_{1.59} = 0.516$. This indicates that under the most adverse combination of values for the geometric parameters explored, and based on the assumptions adopted, a cob wall would have to be subjected to an acceleration of 4.00 m/s$^2$ or 0.408 g to generate the out-of-plane mechanism if the yield compressive strength value is taken into account, and would have to be subjected to an acceleration of 4.69 m/s$^2$ or 0.478 g to

generate the out-of-plane mechanism if the ultimate compressive strength value is taken into account. As these levels of acceleration correspond to the upper range values for a high-seismic-hazard area, and since Ireland is located in a low-seismic-hazard area, it can be concluded that the in-plane mechanism analyzed would not form in Irish surviving cob walls.

Equations (14) and (15) were also used to create the response surfaces shown in Figure 12 for values of compressive strength of 0.48 MPa and 1.59 MPa. The response surfaces show information regarding the effect of two input parameters, wall height and length, on the response, $\alpha$. These response surfaces reflect the quadratic effect the input parameters have on the response.

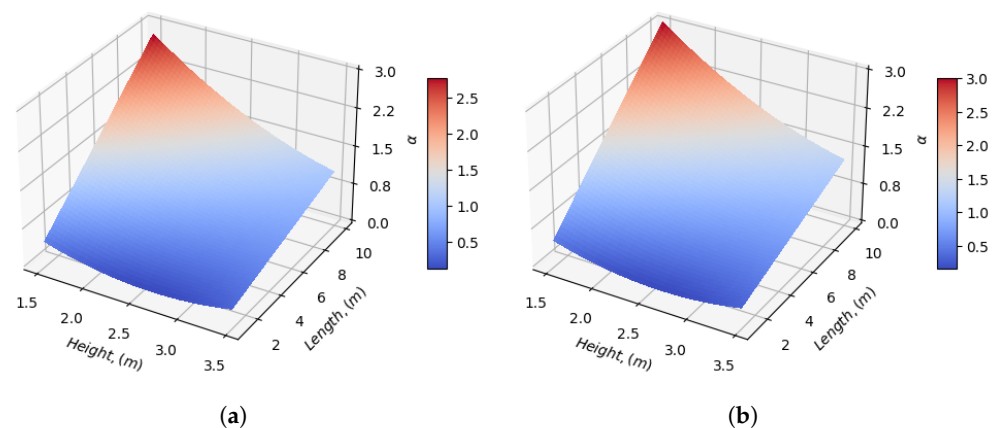

(**a**)                                 (**b**)

**Figure 12.** Response surface for the in-plane mechanism for a compressive strength of cob of (**a**) 0.48 MPa and (**b**) 1.59 MPa.

### 3.7. Key Numerical Findings Presentation and Discussion

The results of these macro-element in-plane analyses are in agreement with the results obtained with the FEM in-plane analysis. The expected negligible influence of the wall thickness on the response was confirmed by both analysis approaches. Furthermore, both analyses' results demonstrated that the cob wall, even under the *"worst-case scenario"* assumptions adopted, would not be damaged when subjected to the range of lateral accelerations expected in Ireland.

Walls are structural elements with higher inertia in their plane than in their out-of-plane direction. Therefore, walls can resist relatively high lateral forces within their plane in comparison to lateral forces out of their plane. The parametric analyses performed and presented in this work are in agreement with this fact. The values presented in the last column of Table 3 show the response values computed using the corresponding equations (and the critical values for the input parameters) for every parametric analysis performed. The safety factors obtained for the in-plane FEM analyses are higher than for the out-of-plane FEM analyses, which indicates that higher acceleration values would be required to cause damage to the wall for the in-plane than for the out-of-plane behavior conditions studied. Similarly, higher values for the in-plane seismic mass multipliers are required to develop the in-plane mechanism than for the development of the out-of-plane mechanism according to the macro-element limit analysis results obtained.

**Table 3.** Critical response values obtained with the different parametric analyses performed.

| Analysis Type | Mechanism | Response | Value |
|---|---|---|---|
| FEM | In-plane | $F_{ys}$ | 5.19 |
| | Out-of-plane | $F_{ys}$ | 1.09 |
| Limit Analysis | In-plane | $\alpha$ | 0.497 |
| | | $\alpha_{0.48}$ | 0.376 |
| | | $\alpha_{1.59}$ | 0.516 |
| | Out-of-plane | $\alpha$ | 0.123 |
| | | $\alpha_{0.48}$ | 0.080 |
| | | $\alpha_{1.59}$ | 0.130 |

## 4. Conclusions and Future Directions

The results obtained with the macro-element models are in agreement with those of the FEM analyses. The expected negligible influence of the wall length on the wall's out-of-plane response was confirmed by both analysis approaches. Similarly, the expected negligible influence of the wall thickness on the wall's in-plane response was also confirmed by both analysis approaches. Furthermore, both analyses results demonstrated that the cob wall, even under the *"worst-case scenario"* assumptions adopted, would not be damaged when subjected to the range of lateral accelerations expected.

Based on the results obtained with both simplified analyses, traditional cob walls in Ireland could be described structurally as very robust. It was observed that relatively high acceleration values, unlikely to happen in a low-seismic-hazard region such as Ireland, would be needed to start the collapse mechanisms studied or cause yielding in typical vernacular cob walls. Due to the characteristics of the vernacular architecture typologies in the island (single-story buildings with regular rectangular plans), low-seismic-hazard levels, and relatively thick walls, needed in principle for construction purposes, it can be concluded that cob walls would very rarely fail under the seismic loads that they may normally be subjected to. Proof of this fact is the many remaining cob buildings which, providing they had been adequately protected against rain and rising damp, have survived for hundreds of years in a relatively good state.

The equations generated with the refined regression models, based solely on the geometry of the walls and acceleration values, can be used by practitioners as a first approach to estimate the safety levels of existing cob buildings in Ireland as well as in other countries where buildings with similar characteristics exist, such as in the United Kingdom and the north of France. Nevertheless, it must be borne in mind that the assumptions adopted and the analyses performed to obtain them are conservative. If their application results in the design of an over-invasive intervention, it is advisable to carry out a more sophisticated analysis to try to preserve, as much as possible, the value of the structure and its original fabric.

To put into perspective the work presented in this paper, it may be useful to compare the structural performance of cob walls against masonry walls, which is a more common and better understood typology. Whereas both cob and masonry walls have been used for centuries as traditional building materials and construction techniques, their structural response mainly differs in terms of strength and flexibility. While masonry can achieve higher levels of compressive strength, it usually presents a brittle behavior and cracks appear along its weak planes, i.e., joint locations. On the other hand, cob walls, by being monolithic elements, and thanks to the added tensile strength provided by the added fibers, normally present a more ductile behavior and greater flexibility than masonry walls. These observations are mainly based on the experimental comparison performed between different earthen construction typologies [13].

The logical line of extension for the results presented in this work would consist of the study of different conditions that would affect existing cob walls' responses, i.e., voids, lateral and top movement restrictions, wind load, etc. Furthermore, the simple methodology adopted to generate the parametric analysis presented in this paper could be

also used as part of a more complex seismic vulnerability analysis in countries located in moderate- or high-seismic-hazard areas, as it provides an efficient and relatively fast way to obtain an estimate of the building's structural response.

**Funding:** The APC was funded by Oslo Metropolitan University.

**Institutional Review Board Statement:** Not applicable.

**Informed Consent Statement:** Not applicable.

**Data Availability Statement:** The data presented in this study are available on request from the corresponding author.

**Conflicts of Interest:** The author declares no conflict of interest.

## Abbreviations

The following abbreviations are used in this manuscript:

| | |
|---|---|
| FEM | Finite element method |
| NIAH | National Inventory of Architectural Heritage |
| SHARE | Seismic Hazard Harmonization in Europe |
| PGA | Peak ground acceleration |
| NIKER | Earthquake-induced risk |
| DOEs | Design of experiments |
| CCD | Central composite design |
| ANOVA | Analysis of variance |

## Appendix A. Sensitivity Analysis

A sensitivity analysis is presented in this Appendix to better understand how the uncertainties inherent to the value of the input parameters studied may affect the parametric equations derived and proposed in this work. This analysis also serves as a first step towards the validation of such parametric equations.

The linear elastic structural response of the studied cob walls mainly depends on the values adopted for the Young's modulus of the material. Therefore, a sensitivity analysis is performed accounting for uncertainty in the adopted value of cob's Young's modulus, corresponding to $\pm 10\%$ and $\pm 20\%$. The parametric analysis is performed and the corresponding parametric equations are derived and subsequently evaluated, accounting for the worst-case scenario of the studied cob walls (a wall height of 3.05 m, a thickness of 0.40 m, and an acceleration of 0.981 m/s$^2$).

By assuming an infinite compressive strength of the material, the kinematic equations derived to compute the collapse multiplier $\alpha$ for either the in-plane or out-of-plane cases depend exclusively on geometric parameters, as can be seen in Equations (2) and (5), respectively. As the values of the geometrical parameters was already studied in the DOE, no further sensitivity analysis is performed for the cases where infinite compressive strength has been assumed.

On the other hand, as presented in Equations (3) and (6), the value of $t$ is inversely proportional to the estimated compressive strength of the material, $f_c$, which affects, respectively, the collapse multiplier values of the in-plane and out-of-plane cases. Therefore, a sensitivity analysis is performed accounting for uncertainty in the adopted value of cob's compressive strength corresponding to $\pm 10\%$ and $\pm 20\%$. The parametric analysis is performed and the corresponding parametric equations are derived and subsequently evaluated, accounting for the worst-case scenario geometry of the studied cob walls (a wall height of 3.05 m, a thickness of 0.40 m, and a length of 3.0 m).

The obtained yielding safety factors and collapse multipliers are compared against the reference values estimated in this manuscript and the results are summarized in Table A1.

**Table A1.** Comparison of values obtained after evaluation of parametric equations from the sensitivity analysis.

| Case | Uncertainty | Parametric Equation Evaluation | Comparison to Reference Value |
|---|---|---|---|
| FEM OOP | −20 | 0.8773 | −19.77 |
| | −10 | 0.9924 | −9.24 |
| | 0 * | 1.0934 | 0.00 |
| | 10 | 1.1866 | 8.52 |
| | 20 | 1.3141 | 20.18 |
| FEM IP | −20 | 4.1603 | −19.93 |
| | −10 | 4.6731 | −10.06 |
| | 0 * | 5.1960 | 0.00 |
| | 10 | 5.7191 | 10.07 |
| | 20 | 6.2308 | 19.91 |
| LA OOP | −20 | 0.0654 | −18.95 |
| | −20 | 0.1288 | −0.88 |
| | −10 | 0.0741 | −8.13 |
| | −10 | 0.1297 | −0.17 |
| | 0 * | 0.0807 | 0.00 |
| | 0 * | 0.1299 | 0.00 |
| | 10 | 0.0859 | 6.49 |
| | 10 | 0.1304 | 0.37 |
| | 20 | 0.0902 | 11.81 |
| | 20 | 0.1305 | 0.44 |
| LA IP | −20 | 0.3315 | −9.53 |
| | −20 | 0.5145 | 1.59 |
| | −10 | 0.3543 | −3.31 |
| | −10 | 0.5133 | 1.36 |
| | 0 * | 0.3664 | 0.00 |
| | 0 * | 0.5064 | 0.00 |
| | 10 | 0.3872 | 5.66 |
| | 10 | 0.5122 | 1.13 |
| | 20 | 0.3971 | 8.37 |
| | 20 | 0.5111 | 0.92 |

* Reference values for each case.

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
