# Peer review of "Learning from the Past: Parametric Analysis of Cob Walls"

_applsci, doi:10.3390/app13159045_

Round 1

Reviewer 1 Report

While the topic is promising, there are several significant aspects that need to be addressed to improve the quality and suitability of the paper for publication. I would like to offer you the opportunity to revise the paper and resubmit it for further consideration. Below are the key points that require attention:

1.     Remove Trivial Statements: Eliminate statements such as "We have much to learn from historic buildings," as they are not suitable for scientific papers and do not contribute to the core objective of the paper.

2.     Clarify the Paper's Objective: Clearly state the main objective of the paper and compare it to the existing state-of-the-art. Additionally, expand the references to include papers that deal with traditional masonry structures. For example:

o   Ramírez Eudave, R., Ferreira, T. M., Vicente, R., Lourenco, P. B., & Peña, F. (2023). Parametric and Machine Learning-Based Analysis of the Seismic Vulnerability of Adobe Historical Buildings Damaged After the September 2017 Mexico Earthquakes. International Journal of Architectural Heritage, 1-24.

o   Khan, N. A., Aloisio, A., Monti, G., Nuti, C., & Briseghella, B. (2023). Experimental characterization and empirical strength prediction of Pakistani brick masonry walls. Journal of Building Engineering, 71, 106451.

3.     Improve Figure Resolution: Enhance the resolution of Figure 5 and ensure that all figures are sufficiently informative. If a figure does not contribute significantly, consider either improving it or removing it from the paper.

4.     Equation Refinement: Evaluate the need for Equation 3 and remove any trivial equations that do not add value to the research.

5.     Figure Formatting: Figures 9-26 need improvement. Remove any unnecessary background and save each figure separately, including subfigure labels (a), (b), etc., in the same format as the text. Ensure the figures provide sufficient information and contribute effectively to the research.

6.     Ansys Simulation: Clarify the purpose of running a linear kinematic analysis with Ansys, especially regarding the analytical expression of load multipliers. The section should be presented in a clear and coherent manner. If the analytical expression are available what is the need for ansys?

7.     Calibration and Validation: If you intend to calibrate an empirical equation for the load multiplier, ensure proper random sampling of the parameter space and validate the equations adequately to support your findings.

8.     Comparative Analysis: After conducting the parametric analysis, provide a clear and well-structured conclusion. Include a comparison of cobwall performance against typical masonry walls to understand performance differences effectively.

9.     Material Uncertainty: Address the issue of material uncertainty and discuss how it can be handled in a deterministic manner for accurate results.

10.  Focus on Relevant Outcomes: Eliminate unnecessary plots and focus on presenting more relevant outcomes that contribute novelty to the research.

Author Response

While the topic is promising, there are several significant aspects that need to be addressed to improve the quality and suitability of the paper for publication. I would like to offer you the opportunity to revise the paper and resubmit it for further consideration. Below are the key points that require attention:

Thank you very much for your insightful feedback. All comments have been addressed accordingly. Additions and modifications to the manuscript are indicated with blue font in the revised version of the manuscript to simplify the revision. Overall, I believe that the quality of the manuscript has improved a lot thanks to the work done based on your comments and suggestions.

  1. Remove Trivial Statements: Eliminate statements such as "We have much to learn from historic buildings," as they are not suitable for scientific papers and do not contribute to the core objective of the paper.

The phrase has been removed both from the abstract and from the introduction as suggested.

  1. Clarify the Paper's Objective: Clearly state the main objective of the paper and compare it to the existing state-of-the-art. Additionally, expand the references to include papers that deal with traditional masonry structures. For example:
  • Ramírez Eudave, R., Ferreira, T. M., Vicente, R., Lourenco, P. B., & Peña, F. (2023). Parametric and Machine Learning-Based Analysis of the Seismic Vulnerability of Adobe Historical Buildings Damaged After the September 2017 Mexico Earthquakes. International Journal of Architectural Heritage, 1-24.
  • Khan, N. A., Aloisio, A., Monti, G., Nuti, C., & Briseghella, B. (2023). Experimental characterization and empirical strength prediction of Pakistani brick masonry walls. Journal of Building Engineering, 71, 106451.

As suggested, the objective of the paper has been restated as follows:

The objective of this paper is to analyze existing cob vernacular buildings to provide simplified analysis tools for practitioners to better understand their structural response. To achieve this, the influence that geometric parameters and external actions have in the response of cob walls has been identified by the means of a parametric analysis. Guidance is provided in the form of parametric equations capable of computing collapse multipliers (for the limit analysis approach) or safety factors (for the linear elastic Finite Element Method (FEM) approach). Furthermore, such guidance could as well be used for the design of new sustainable and resilient cob buildings, which is a field that have recently received increased attention [15–19].

Moreover, the following paragraphs and complementary references (including the two suggested references) have been added to enrich the presentation of the different earthen structure typologies and the structural differences among them:

Earthen vernacular architecture structures can be classified in different ways based on i) the construction method type as either dry or wet, ii) the structural function as load bearing or non-load bearing, and iii) the structural element type as monolithic, masonry or infill. Examples of masonry earthen structures are adobe [5], compressed earth blocks [6], and sod/turf [7]. On the other hand, wattle-and-daub [8] and shot earth [9] are examples of infill earthen structures, whereas that rammed earth [10] and cob [11] elements are normally classified as monolithic structural elements. Infill elements are normally used as partition walls and act as non-load bearing elements, on the contrary, both masonry and monolithic elements normally support their own weight as well as other parts of the structure acting as load bearing elements. As recognized by [12], masonry is a non-homogeneous anisotropic material and its correct characterization need to account for the inherent properties of its units (adobes, compressed earth blocks, etc.) and joints (mortar or the lack of it). Conversely, earthen monolithic elements can be idealized as homogeneous, in analogy to the manner in which concrete is normally understood, and their characterization is performed based on the bulk properties of the material [13].

Moreover, simplified analysis tools such as the one proposed in this paper are usually seek after to be implemented in vulnerability assessment of a large number of built assets located within a certain region. Vulnerability assessments of masonry building [20] and adobe building [21] typologies are available in the literature. On the other hand, to the extent of the knowledge of the author of this paper, no such study has been performed yet for cob vernacular buildings in Ireland or in other regions with similar building typologies. Thus, the work presented in this manuscript represents a novel contribution to the field as a first attempt to investigate the performance of this group of built assets.

  1. Improve Figure Resolution: Enhance the resolution of Figure 5 and ensure that all figures are sufficiently informative. If a figure does not contribute significantly, consider either improving it or removing it from the paper.

It was decided to remove the figure as it did not contribute significantly, as per suggested. Other figures related to the main and interaction effect plots of the studied parameters have as well been removed for similar reasons.

  1. Equation Refinement: Evaluate the need for Equation 3 and remove any trivial equations that do not add value to the research.

The equation has been removed from the manuscript as suggested and a short description within the paragraph was added instead. Moreover, other non-essential equations have as well been removed (namely, equations number 2, 4 and 11).

  1. Figure Formatting: Figures 9-26 need improvement. Remove any unnecessary background and save each figure separately, including subfigure labels (a), (b), etc., in the same format as the text. Ensure the figures provide sufficient information and contribute effectively to the research.

As previously mentioned, figures related to the main and interaction effect plots of the studied parameters have been removed. The quality of all response surface figures has been improved, unnecessary background removed, and each figure has been saved separately as requested.

  1. Ansys Simulation: Clarify the purpose of running a linear kinematic analysis with Ansys, especially regarding the analytical expression of load multipliers. The section should be presented in a clear and coherent manner. If the analytical expression are available what is the need for ansys?

It seems to me that there has been a confusion here. ANSYS was used to perform linear elastic FEM analysis, not linear kinematic analysis. These analyses were needed to compute the safety factors values, which were chosen as the response parameters of the parametric analysis.

  1. Calibration and Validation: If you intend to calibrate an empirical equation for the load multiplier, ensure proper random sampling of the parameter space and validate the equations adequately to support your findings.

This is a concern also point out by the other reviewer. To address both comments, a new annex has been added to the manuscript where the results of a sensitivity analysis are presented to better understand the effect that uncertainties present in the estimation of the material parameters have in the proposed equations and validate their use.

  1. Comparative Analysis: After conducting the parametric analysis, provide a clear and well-structured conclusion. Include a comparison of cobwall performance against typical masonry walls to understand performance differences effectively.

The length of the conclusions has been reduced, also as it was a request done by the other reviewer, and the following paragraph was added to discuss a little bit (due to length restrictions) about the differences on the structural performance of cob walls against masonry walls:

To put into perspective the work presented in this paper, it may be useful to compare the structural performance of cob walls against masonry walls, which is a more common and better understood typology. Whereas both cob and masonry walls have been used for centuries as traditional building materials and construction techniques, their structural response mainly differs in terms of strength and flexibility. While masonry can achieve higher levels of compressive strength, it usually presents a brittle behavior and cracks appear along its weak planes, i.e., joint locations. On the other hand, cob walls, by being monolithic elements and thanks to the added tensile strength provided by the added fibers, normally present, a more ductile behavior and greater flexibility than masonry walls. These observations are mainly based on the experimental comparison performed between different earthen construction typologies [13].

  1. Material Uncertainty: Address the issue of material uncertainty and discuss how it can be handled in a deterministic manner for accurate results.

Material uncertainty has been further discussed in the newly added annex of the manuscript.

  1. Focus on Relevant Outcomes: Eliminate unnecessary plots and focus on presenting more relevant outcomes that contribute novelty to the research.

As suggested and as previously mentioned in the replies of previous points, unnecessary plots have been removed and only those presented relevant outcomes have remained.

Reviewer 2 Report

The below issues must be addressed

1-add some numerical key findings

2-add limitation of the study

3-validity of the test/model test results must be provided in a separate supplementary file

4-conclusion must be shortened

the paper is well written there are some minor English grammar issues

Author Response

The below issues must be addressed

Thank you very much for your insightful feedback. All comments have been addressed accordingly. Additions and modifications to the manuscript are indicated with blue font in the revised version of the manuscript to simplify the revision. Overall, I believe that the quality of the manuscript has improved a lot thanks to the work done based on your comments and suggestions.

1-add some numerical key findings

A subsection header at the end of the Results section was added to clearly indicate the discussion about the numerical key findings of the presented work, which are summarized in Table 3.

2-add limitation of the study

The following subsection has been added at the end of the Methodology section to highlight the limitations of the study:

Study limitations

The main limitations of the current study are related to the inherent numerical modeling simplifications adopted and to the uncertainties arising from the selections of parameter values due to the scarce availability of such information. When a simplified linear elastic analysis of an existing cob structure cannot be justified, either because future seismic events, high loading levels or other conditions may be expected to cause a non-linear response of the structure, the use of more advanced numerical modeling tools, such as the ones discussed in [51], is recommended.

Unfortunately, cob buildings have not received as much attention as other construction typologies, i.e., adobe or masonry, and as discussed in this section, the values of the parameters selected are based on limited experimental laboratory campaigns and registries in historical sources. Although the parametric equations proposed in this work have been derived using well known and accepted principles [52, 53], for a robust and strict validation, further work is required. In Annex A a sensitivity analysis is presented to better understand how the uncertainties of key material parameters influence the parametric equations proposed in this work and as a step forward towards their validation.

3-validity of the test/model test results must be provided in a separate supplementary file

This is a concern also point out by the other reviewer. To address both comments, a new annex has been added to the manuscript where the results of a sensitivity analysis are presented as a means to better understand the effect that uncertainties present in the estimation of the material parameters have in the proposed equations and validate their use.

4-conclusion must be shortened

Conclusions have been shortened as suggested by removing a few paragraphs as they contained non-essential information. On the other hand, a new paragraph has been added as per request of the other reviewer to discuss a little bit about how cob and masonry walls compare with each other.

Furthermore, the whole manuscript has been reviewed using the Grammarly tool and minor editing and English corrections have been performed, as indicated.

Round 2

Reviewer 1 Report

Accept